# Electromagnon dispersion probed by inelastic X-ray scattering in LiCrO$_2$

Sándor Tóth[1,2], Björn Wehinger[1,3], Katharina Rolfs[4], Turan Birol[5,6], Uwe Stuhr[1], Hiroshi Takatsu[7,8], Kenta Kimura[9], Tsuyoshi Kimura[9], Henrik M. Rønnow[2] & Christian Rüegg[1,3]

Inelastic X-ray scattering with meV energy resolution (IXS) is an ideal tool to measure collective excitations in solids and liquids. In non-resonant scattering condition, the cross-section is strongly dominated by lattice vibrations (phonons). However, it is possible to probe additional degrees of freedom such as magnetic fluctuations that are strongly coupled to the phonons. The IXS spectrum of the coupled system contains not only the phonon dispersion but also the so far undetected magnetic correlation function. Here we report the observation of strong magnon–phonon coupling in LiCrO$_2$ that enables the measurement of magnetic correlations throughout the Brillouin zone via IXS. We find electromagnon excitations and electric dipole active two-magnon excitations in the magnetically ordered phase and heavily damped electromagnons in the paramagnetic phase of LiCrO$_2$. We predict that several (frustrated) magnets with dominant direct exchange and non-collinear magnetism show surprisingly large IXS cross-section for magnons and multi-magnon processes.

[1] Laboratory for Neutron Scattering and Imaging, Paul Scherrer Institute, 5232 Villigen, Switzerland. [2] Laboratory for Quantum Magnetism, Institute of Physics, EPFL, 1015 Lausanne, Switzerland. [3] Department of Quantum Matter Physics, University of Geneva, 1211 Genève, Switzerland. [4] Laboratory for Scientific Developments and Novel Materials, Paul Scherrer Institute, 5232 Villigen, Switzerland. [5] Department of Chemical Engineering and Materials Science, University of Minnesota, Minneapolis, Minnesota 55455, USA. [6] Department of Physics and Astronomy, Rutgers University, Piscataway, New Jersey 08854, USA. [7] Department of Energy and Hydrocarbon Chemistry, Graduate School of Engineering, Kyoto University, Kyoto 615-8510, Japan. [8] Department of Physics, Tokyo Metropolitan University, Tokyo 192-0397, Japan. [9] Division of Materials Physics, Graduate School of Engineering Science, Osaka University, Osaka 560-8531, Japan. Correspondence and requests for materials should be addressed to S.T. (email: sandor.toth@psi.ch).

The coupling between magnetic and lattice degrees of freedom gives rise to many interesting effects. It can induce multiferroic order with ferroelectric polarization coupled to the magnetic structure[1–3] or it can generate dynamic mixed magnon–phonon excitations. If the magnon is coupled to a polar phonon, the mixed mode, termed electromagnon, can be excited by the electric field of light at the resonant frequency[4–6]. Previous experiments showed that magnetization dynamics of materials can be studied at ultrafast timescales by exciting electromagnons via femtosecond light pulses[7]. Moreover, optical properties of magnetoelectric materials at the resonant frequency can be controlled via external magnetic field[8,9]. Measurement of electromagnons is possible via THz spectroscopy. However, this technique is able to probe only the centre of the Brillouin zone. As we shall show, electromagnons can appear at finite momentum, thus inaccessible to THz spectroscopy. Inelastic neutron scattering can also identify the magnetic and phononic component of an electromagnon excitation, however previous studies found only small energy shifts of the magnons due to magnon–phonon coupling[10,11], while the transfer of spectral weight between magnons and phonons could not be resolved so far. Here we show that LiCrO$_2$ is an exceptional material, where the magnon–phonon coupling is strong enough to make the transferred spectral weight from phonons to magnons visible for inelastic X-ray scattering and thus enables the direct measurement of the electromagnon dispersion. We also suggest additional systems, where similarly strong effects can be present.

LiCrO$_2$ is an excellent realization of the two-dimensional (2D) $S = 3/2$ Heisenberg triangular lattice antiferromagnet (TLA) model with only minimal corrections due to structure and symmetry. Dzyaloshinskii–Moriya interactions are forbidden on all bonds due to the space group symmetry of $R\bar{3}m$. Single-ion anisotropy is expected to be small due to the octahedral coordination of the Cr$^{3+}$ ion, which have half-filled $t_{2g}$ shells resulting in quenched orbital angular momentum. The interplane interactions are weak due to the large separation of the triangular layers. LiCrO$_2$ develops long-range magnetic order at $T_N = 61.2$ K (ref. 12). The magnetic structure is an $ac$-plane helical order with wavevectors of $\mathbf{k}_m = (1/3, 1/3, 0)$ and $\mathbf{k}_m = (-2/3, 1/3, 1/2)$. The angles between neighbouring spins on the triangular planes are exactly 120° and the chirality is staggered along the $c$ axis as a result of the double-$Q$ structure[13]. The staggered chirality implies the appearance of a small symmetry breaking term in the spin Hamiltonian below $T_N$. The magnetic interactions in the plane are dominated by direct exchange[14]. These interactions are sensitive to the modulation of the bond length, similarly to other Cr$^{3+}$ compounds with short bonds such as ZnCr$_2$O$_4$ (refs 15,16) and MgCr$_2$O$_4$ (ref. 17). Furthermore, LiCrO$_2$ shows a pronounced anomaly in the dielectric constant at $T_N$ but no ferroelectric polarization could be observed[18] pointing towards an antiferroelectric ground state induced by the staggered chirality of the triangular layers[19,20].

We report the spin and lattice excitation spectrum of LiCrO$_2$ measured via inelastic X-ray scattering with meV energy resolution (IXS) and inelastic neutron scattering (INS), and a direct observation of an electromagnon. The data reveal a surprisingly strong mixing between magnons and phonons. We show that the observed electromagnon is the result of the strong coupling between the phason spin wave mode and the longitudinal acoustic (LA) phonon in the magnetically ordered phase of LiCrO$_2$. We also present a model that describes both the measured quasiparticle dispersion and the cross-sections for IXS and INS at low temperature. This model shows that non-collinear magnetic order and exchange striction (ES) can induce linear coupling between magnons and phonons. Furthermore, we report the observation of two-magnon (2M) excitations in the

ordered phase and strongly damped excitations above the Néel temperature from our IXS data.

## Results

**Temperature-dependent IXS spectrum of LiCrO$_2$.** The IXS spectrum of LiCrO$_2$ measured at room temperature at $Q = (1.5, 1.5, 0)$ shows three phonon modes at energies of 30.8(4), 34.6(2) and 59.2(1) meV (Fig. 1 and Supplementary Fig. 1). The measured $Q$-point is equivalent to the $M'$-point of the Brillouin zone shifted by $(1, 1, 0)$ (Fig. 2b). The lowest-energy mode has an unusually large intrinsic width of 6.8(5) meV (see Methods for details of the data analysis). On cooling, the phonon spectrum goes through a marked change. The lowest phonon peak loses almost all of its intensity and a new resonance develops gradually below 10 meV. This new mode appears as a broad diffuse scattering signal at 99 K, centred at about 8 meV. With decreasing temperature, the peak becomes more pronounced accompanied with increasing spectral weight and decreasing width. At the lowest measured temperature of 7 K the peak position is at

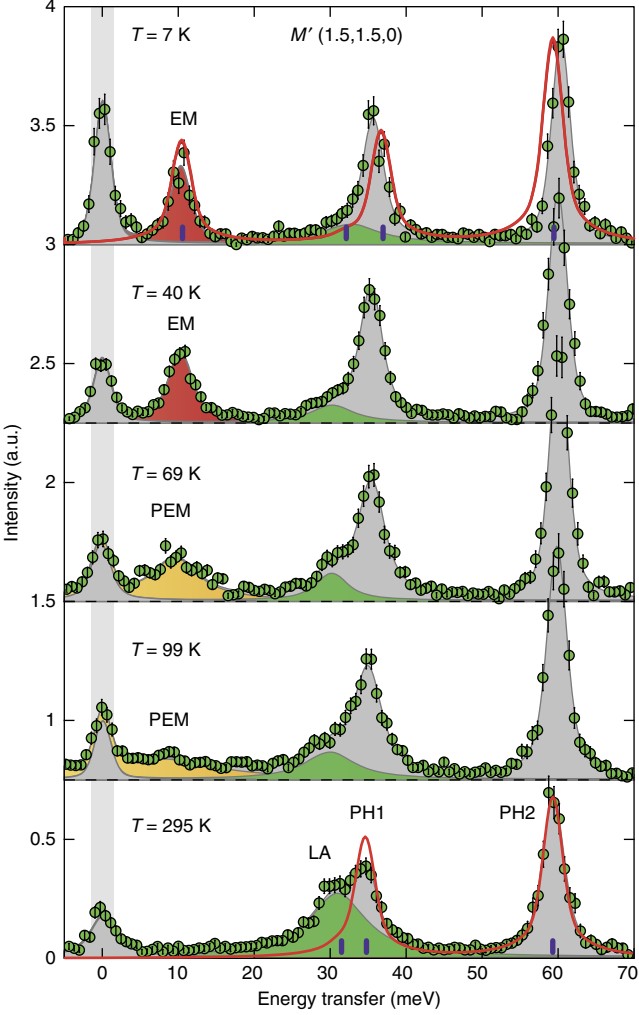

**Figure 1 | Temperature evolution of the measured IXS spectrum at the $M'$-point.** Green dots denote experimental data normalized to the same monitor. Red, yellow, green and grey filled areas are fitted electromagnon (EM), paraelectromagnon (PEM), LA and optical phonon peaks (PH1 and PH2), respectively. Light grey area shows the full width at half maximum of the elastic line. Red lines are the theoretical spectrum of the coupled model at 7 K and the pure phonon model at 295 K. Vertical purple lines show the calculated quasiparticle energies. Error bars indicate 1 s.d.

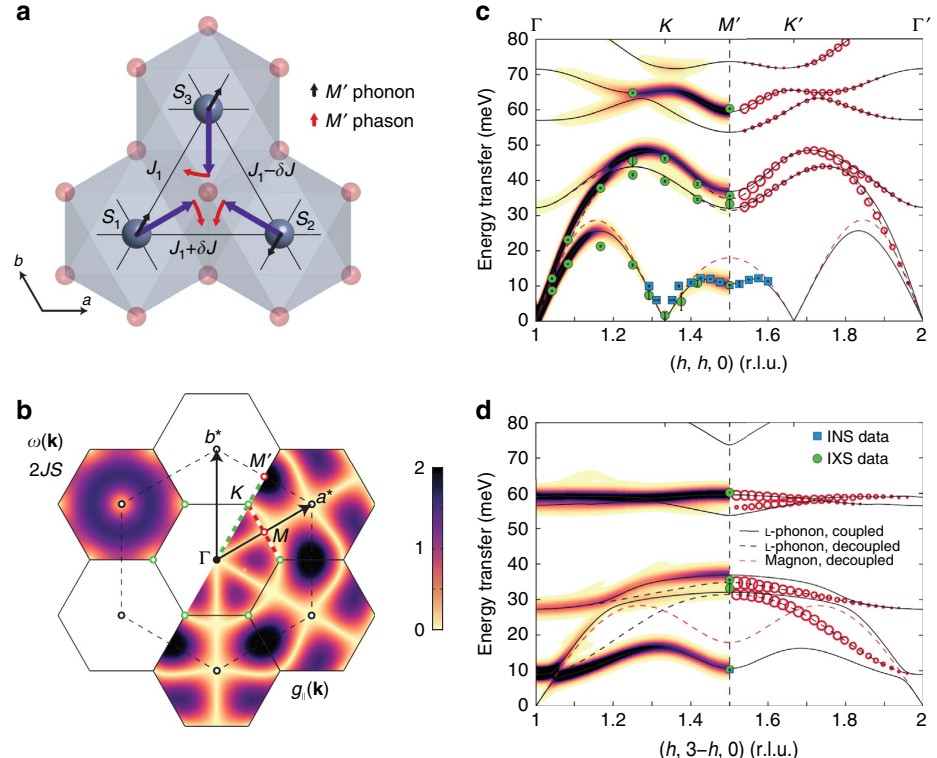

**Figure 2 | Coupled magnon and phonon modes in LiCrO₂.** (**a**) A single triangle of $Cr^{3+}$ spins is shown with the surrounding $O_6$ octahedra. Purple arrows depict the helical magnetic structure (rotated into the $ab$-plane for better visibility), the phonon and phason amplitude at the $M'$-point is shown by black and red arrows, respectively. (**b**) Reciprocal space of the triangular lattice with black and dashed hexagons denoting the magnetic and crystallographic Brillouin zones, respectively. The upper-left and lower-right colour maps show the phason energy and the $g_{||}(\mathbf{k})$ value, respectively (see text). Green and red dashed lines show the path of the IXS and INS measurement, respectively. (**c**) Comparison of the measured phonon dispersion at 7 K and the coupled magnon–phonon model along the $(h, h, 0)$ direction. The colour map on the left half shows the calculated IXS cross-section in a.u., while the filled green circles and blue squares denote the measured quasiparticle energies using IXS and INS, respectively. The black dashed and red dashed lines denote the magnon and longitudinal phonon dispersion of the uncoupled model, while the continuous black lines correspond to the coupled dispersion. The empty red circles on the right half denote the $\mathbf{e}_\lambda \cdot \mathbf{g}(\mathbf{k})$ factor that determines the strength of the magnon–phonon coupling for each $\lambda$ phonon mode. (**d**) Model calculation for in-plane direction perpendicular to $(h, h, 0)$. Error bars indicate 1 s.d.

10.3(2) meV and it has a resolution limited width. To compare directly the integrated intensities at different temperatures we calculated the dynamical susceptibility for the lowest phonon peak and the new resonance, shown in Fig. 3. It is clearly visible that the lowest energy phonon transfers most of its spectral weight to the new mode on cooling while the sum of the two is temperature independent. The measured room-temperature phonon energies are well reproduced by our *ab initio* calculation (see Methods) shown as vertical purple lines in Fig. 1. The calculated dispersion reveals that the observed phonon with lowest energy is a LA branch while the two peaks at higher energy correspond to optical branches (PH1 and PH2). While the relative spectral weight of the two upper modes is well reproduced by the calculation, the intensity of the LA phonon is strongly underestimated (see red curve in Fig. 1). The calculated energy of the LA phonon is 31.4 meV and its symmetry belongs to the $E_u$ polar representation.

**Electromagnon dispersion in LiCrO₂.** To unambiguously identify the new low-energy mode of LiCrO₂ we measured the excitation spectrum by INS along the $(h, 1 - 2h, 0)$ reciprocal space direction equivalent to $(h, h, 0)$ in the magnetic Brillouin zone (Supplementary Fig. 2). At the $M$-point (equivalent to $M'$) a single spin wave excitation was found in the helical phase at 1.5 K centred at 10.3(1) meV. Since neutrons are sensitive to magnetic

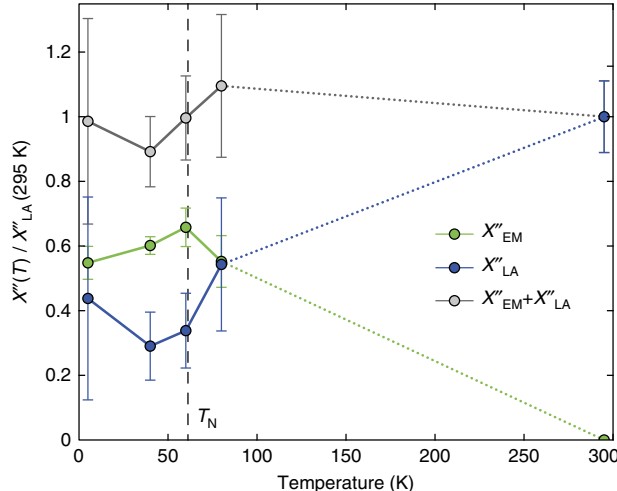

**Figure 3 | Imaginary part of the dynamical susceptibility at the $M'$-point.** The green and blue dots denote the integrated signal of the electromagnon (EM) and LA phonon after correction for the Bose factor. All values are normalized to the room temperature value of the LA phonon. Error bars indicate 1 s.d.

fluctuations in the measured momentum range, we can conclude that the low-energy resonance at $T < T_N$ has not only polar phononic but also magnetic character, thus it is an electromagnon with a finite momentum. Moreover, at intermediate temperatures above $T_N$ the strongly damped low-energy excitation might be a paraelectromagnon due to the lack of both magnetic and electric dipole order. The origin of this excitation can be a phonon coupled to the excitation of the 2D correlated magnetic state, which persists above $T_N$ due to the low dimensionality of the system[13,21,22].

**Two-magnon continuum.** To determine the coupling mechanism that drives the observed strong magnon–phonon mixing, we measured IXS spectra at multiple points along the $(h, h, 0)$ reciprocal space direction at 7 K and fitted the phonon energies. The electromagnon spectrum is reported in Fig. 4a–i (also in Supplementary Fig. 3). Remarkably, the energy width of the

electromagnon excitation increases substantially around the magnetic Bragg point (K-point). Since the one magnon excitations are sharp at low temperature, the broad IXS peaks can be due to phonons coupled to the 2M continuum that is intrinsically broad for dispersive magnons. The 2M continuum is a purely quantum effect and the corresponding dynamical structure factor is typically much smaller than that of the single magnon. It is related to the longitudinal fluctuations of the ordered spins[23]. To corroborate our argument, we calculated the non-interacting 2M dynamical structure factor for the TLA (ref. 23) with first- and second-neighbour antiferromagnetic interactions $J_1 = 8.17$ meV and $J_2 = 0.556$ meV, shown in Fig. 4j; and a cut at $(1.292, 1.292, 0)$ reciprocal space point is shown in Fig. 4e. The 2M dynamical structure factor is strongest close to the K-point, and the centre of the 2M spectral weight is expected to be close to the one-magnon energy. This can explain why the measured electromagnon spectrum continuously changes from a sharp one magnon–one phonon mode to a phonon mixed with the 2M

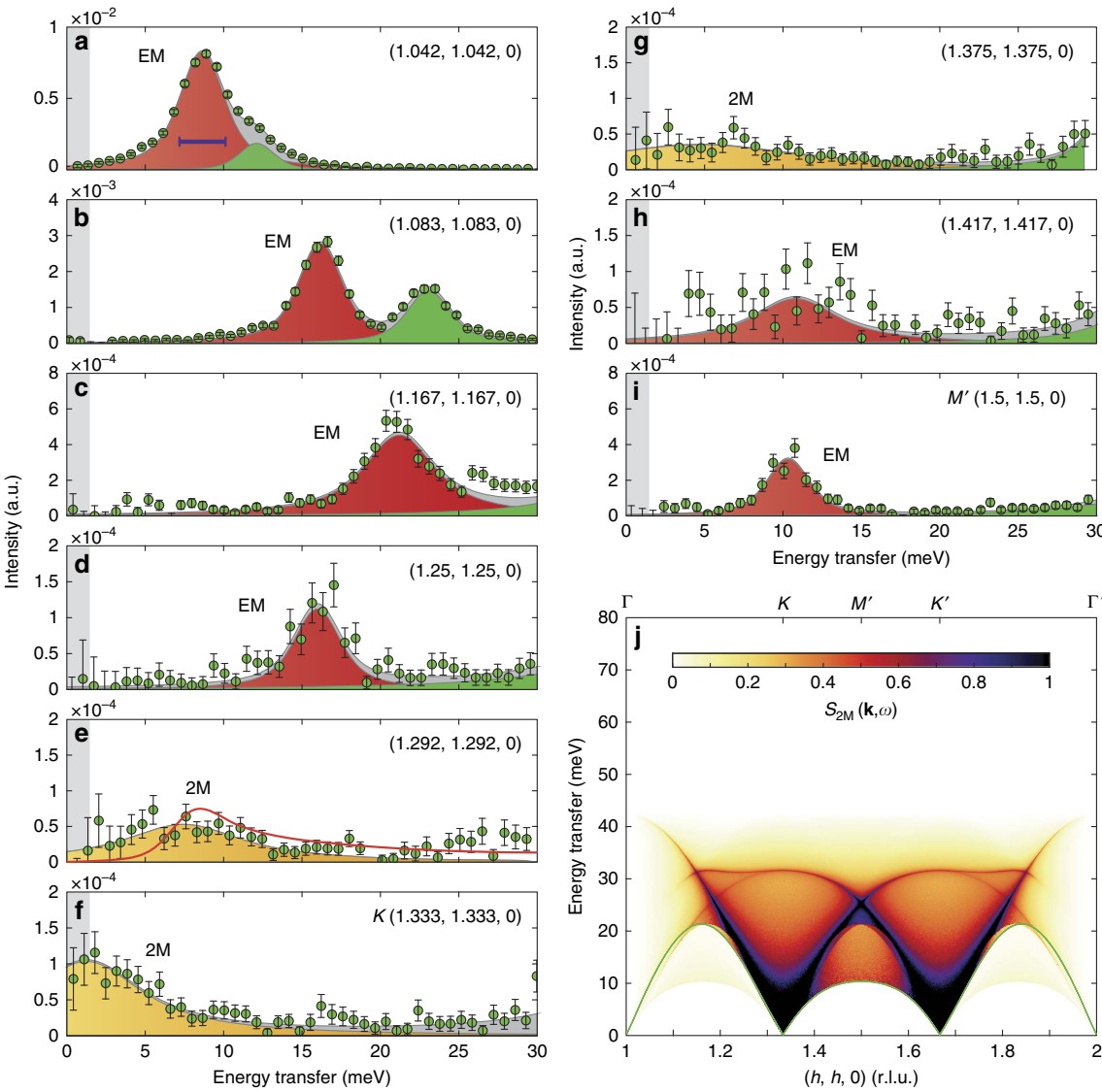

**Figure 4 | IXS electromagnon spectrum measured at 7 K and the 2M continuum.** (**a–i**) Green dots denote experimental data along $(h, h, 0)$ normalized to the same monitor after the subtraction of the elastic peak. Red, yellow, green and grey filled areas are fitted electromagnon (EM), 2M, LA and optical phonon peaks (PH1 and PH2), respectively. Light grey area shows the full width at half maximum (FWHM) of the elastic line. The plots are scaled individually to enhance the visibility of the weak peaks. The red line in **e** shows the theoretical 2M spectrum convoluted with the experimental resolution function at $Q = (1.292, 1.292, 0)$. The horizontal purple bar in **a** shows the FWHM of the instrumental resolution function. (**j**) The absolute value of the 2M dynamical structure function for the pure magnon model. Green line denotes the dispersion of the phason spin wave mode. Error bars indicate 1 s.d.

continuum as its momentum gets closer to the $K$-point. In the following we model only the single magnon–phonon spectrum.

**Phason spin wave mode as electromagnon.** The fitted peak positions of both the INS and IXS data are presented in Fig. 2c,d (Supplementary Tables 1–3) together with the model calculations, which will be explained in the following. In general, helical magnetic structures have three spin wave modes: a phason mode with rotation of all spins in the ordering plane and two canting modes correspond to spins canting away from the ordering plane. Strikingly, our measured electromagnon spectrum contains only one of the three spin wave modes that according to its dispersion corresponds to the phason mode of the helical magnetic structure. Moreover, the two canting modes of the spin spiral are completely decoupled from the phonons. Besides, the phason mode shows a roton-like minimum at $M'$. Similar minima were previously observed in several TLAs such as $CuCrO_2$ (refs 24,25), $\alpha$-$CaCr_2O_4$ (ref. 26) and $LuMnO_3$ (ref. 27). This points towards a general sensitivity of the magnon energy at the $M$-point to perturbations such as quantum fluctuations[28,29], further neighbour interactions or magnon–phonon coupling[30]. The electromagnon in $LiCrO_2$ has large IXS scattering cross-section at both the $\Gamma$- and $M'$-points.

**Model of the magnon–phonon coupling.** The microscopic mechanism that couples the magnons and phonons in $LiCrO_2$ is the symmetric ES, since the antisymmetric exchange is too weak being a relativistic correction[31]. In the following we will show that the measured electromagnon dispersion and IXS cross-section can be well described on a single triangular layer assuming strong ES between first-neighbour chromium atoms. We will show that in non-collinear magnets ES gives a linear coupling between magnons and phonons thus can generate a strong mixing (for a detailed description see Supplementary Note 1). For a quantitative description, we propose the following Hamiltonian that couples spins to phonons, taking into account the ideal isotropic nature of the spins in $LiCrO_2$:

$$\mathcal{H} = J(r) \sum_{m,n} \mathbf{S}_m \cdot \mathbf{S}_n + \mathcal{H}_L, \qquad (1)$$

where $J(r)$ is the Heisenberg exchange between first-neighbour spins as a function of the bond length $r$, $\mathbf{S}_m$ is the spin vector operator on the $m$th magnetic atom and $\mathcal{H}_L$ is the Hamiltonian of the lattice vibrations. To simplify equation (1), we keep only the constant and linear term from the Taylor expansion of $J(r)$ around the $r_0$ equilibrium bond length. The constant term $J_1$ describes the spin wave dynamics in the absence of phonons, while the linear coefficient $J_{mp}$ gives the leading magnon–phonon coupling term:

$$\mathcal{H}_{mp} = J_{mp} \sum_{m,n} \hat{\mathbf{d}}_{mn} \cdot (\mathbf{u}_m - \mathbf{u}_n) \mathbf{S}_m \cdot \mathbf{S}_n, \qquad (2)$$

where $\mathbf{u}_m$ is the displacement vector of atom $m$ and $\hat{\mathbf{d}}_{mn}$ is the unit bond vector pointing from atom $m$ to atom $n$. In the magnetically ordered phase if the order is non-collinear $\mathcal{H}_{mp}$ linearly couples the phonon and magnon bosonic operators $a_\lambda(\mathbf{k})$ and $b(\mathbf{k})$. After applying the linear Holstein–Primakoff approximation and using a rotating coordinate system for the spins[23,29,32] the equation simplifies to

$$\mathcal{H}_{mp} = i \sum_{\mathbf{k},\lambda} \gamma_\lambda(\mathbf{k}) a_\lambda(\mathbf{k}) \left( b^\dagger(\mathbf{k}) - b(-\mathbf{k}) \right) + h.c., \qquad (3)$$

where $\lambda$ indexes the phonon modes. The coupling term $\gamma_\lambda(\mathbf{k})$ is given by

$$\gamma_\lambda(\mathbf{k}) = -\frac{3}{4} J_{mp} S \sqrt{\frac{S\hbar}{M\omega_\lambda(\mathbf{k})}} \, \mathbf{e}_\lambda(\mathbf{k}) \cdot \mathbf{g}(\mathbf{k}), \qquad (4)$$

where $M$ is the mass of the magnetic atom, $\omega_\lambda(\mathbf{k})$ and $\mathbf{e}_\lambda(\mathbf{k})$ are the energy and amplitude of the $\lambda$ phonon on the chromium atom. The $\mathbf{g}(\mathbf{k})$ geometrical factor for the Bravais lattice of magnetic atoms reads

$$\mathbf{g}(\mathbf{k}) = \sum_{\mathbf{d}} \hat{\mathbf{d}} \sin(2\pi \mathbf{k}_m \cdot \mathbf{d}) [\cos(2\pi \mathbf{k} \cdot \mathbf{d}) - 1], \qquad (5)$$

where the sum runs through bonds denoted by $\mathbf{d}$ (with ES). The linear coupling vanishes for collinear magnetic order, because $\mathbf{g}(\mathbf{k})$ is zero for $\mathbf{k}_m = 0$. It is also zero at the $\mathbf{k} = \mathbf{k}_m$ reciprocal space point, explaining why no one-magnon excitation is visible on the IXS spectrum close to the $K$-point. The coupled model can be solved using Bogoliubov transformation, and the corresponding neutron and X-ray scattering cross-sections can be calculated (Supplementary Note 1). The inactivity of the additional two canting spin wave modes of the helical structure in the IXS spectrum can be explained within our model as follows. The exchange interactions in the system can be thought of as effective magnetic fields acting on each magnetic site and being equal to the sum of the neighbouring moments times the exchange constant $J_1$. In the absence of phonons, the field is parallel to the moment direction on every site as illustrated by purple arrows in Fig. 2a. However, when a phonon perturbs the system and modulates the uniform $J_1$ via ES, the effective magnetic field will not be parallel to the moment direction any more but points somewhere within the plane of the spin spiral. This will induce a modulation of the phase of the spins within the spiral as spins reorient themselves to minimize the total energy. This phase modulation is exactly the phason spin wave mode that we see in our data.

## Discussion

For a full interpretation of the IXS spectrum of $LiCrO_2$, we start with the pure phonon spectrum in the paramagnetic phase determined from *ab initio* calculation. The dispersion of the longitudinal phonons are shown in Fig. 2c,d by black dashed lines and the full phonon spectrum in Supplementary Fig. 4. The calculated dispersion relation agrees well with the measured phonon energies showing that the magnon–phonon coupling introduces only minor energy shifts. The introduction of $J_{mp}$ will mix the phason and phonon amplitudes. The strongest mixing is calculated to be between the LA and transverse acoustic phonon branches of the 2D triangular planes and the phason spin wave mode of the helical magnetic structure in agreement with the experiment. The wavevector-dependent intensity of the IXS electromagnon signal is proportional to $g_{\parallel}(\mathbf{k}) = \mathbf{g}(\mathbf{k}) \cdot \hat{\mathbf{k}}$, which is largest along the $(h, h, 0)$ in reciprocal space and zero at lattice and magnetic Bragg points (Fig. 2b). It is important to note that although $g_{\parallel}(\mathbf{k})$ is zero at the $M$-point, the coupled dispersion is the same as at $M'$ just both $\mathbf{g}(\mathbf{k})$ and $\mathbf{e}(\mathbf{k})$ vectors are rotated by 90° thus invisible for IXS. The largest mixing amplitude is expected at $\Gamma$ and $M'$ in agreement with our experimental results. Remarkably, the strong coupling causes a roton-like minimum of the spin wave dispersion at $M'$ downwards renormalizing the phason energy by 42%, when compared with the decoupled model, even though the lowest phonon mode is 20 meV higher in energy.

To determine the parameters of the coupled model, we fitted the experimental electromagnon dispersions using $J_1$ and $J_{mp}$ as parameters. The best model parameters are $J_1 = 6.00(25)$ meV and $J_{mp} = 65(4)$ meV Å$^{-1}$. Including an additional second-neighbour exchange interaction in the triangular planes results in zero within error bar. The optimized coupled model describes both the measured dispersion (see black lines in Fig. 2c) and the IXS cross section (see red line in Fig. 1) very well. Some deviation close to the $\Gamma$-point is due to the overestimation of the speed of sound from the *ab initio* calculation. The real-space dynamics of the strong coupling at the $M'$-point is visualized in Fig. 2a. At this

wavevector the LA phonon (black arrows) shortens and lengthens the $S_1$–$S_2$ and $S_2$–$S_3$ bonds, respectively. The excited phason mode is in phase with the phonon that makes the $S_1$–$S_2$ bond stronger ($J_1 + \delta J$) while the $S_2$–$S_3$ bond weaker ($J_1 + \delta J$). Thus, a ferromagnetic fluctuation on the longer bond and antiferromagnetic fluctuation on the shorter bond is energetically favourable if ES is present. This explains the reduction of the phason energy and the roton-like minimum. The $S_1$–$S_2$ bond is inactive at this wavevector since it changes neither length nor relative spin orientation. The largest electromagnon cross-section is predicted close to the $\Gamma$-point in agreement with experiment. Although the $g(k)$ coupling term vanishes at $\Gamma$, the decreasing energy separation between the LA phonon and the phason mode overcomes this reduction towards the zone centre.

There are potentially many other magnetic correlated systems where the magnon–phonon coupling is present and matrix elements are allowed by symmetry. However, for a measurable hybridization between magnetic and lattice fluctuations a large coupling is necessary that makes only a few of them suitable for studying magnetism via IXS. For example, $ZnCr_2O_4$ and $MgCr_2O_4$ with pyrochlore structures have ES values comparable to $LiCrO_2$. It is also possible that the observed molecular resonance-like magnetic signal in these systems[33–36] is related to hybridized magnon–phonon modes. Besides, large magnon–phonon coupling is expected for magnetic $5d$ systems, where the extended $d$ orbitals can support large modulation of the superexchange interaction due to ligand vibrations. For example, in $NaOsO_3$ an upward shift of the optical phonon energy by 5 meV was attributed to the onset of magnetic correlations[37]. $5d$ systems are especially promising for IXS studies as the general lack of large single crystals prohibits detailed INS experiments.

In conclusion, we reported inelastic X-ray scattering data on $LiCrO_2$ that revealed a dispersive electromagnon. Our analysis showed that it is the phason mode of the helical spin order coupled to a LA phonon. We identified the exchange striction between first-neighbour chromium ions as the microscopic coupling mechanism. Fitting the model parameters to the measured electromagnon dispersion we could reproduce both the experimental dispersion and the dynamical structure factor for inelastic X-ray scattering. Beside the one-magnon process we also found signature of coupling between the acoustic phonon branches and the 2M continuum around the magnetic Bragg points that can be explained by including higher-order corrections to our linear theory. In the paramagnetic phase we observed a heavily damped electromagnon that might be stabilized by the low-dimensional magnetic correlations of the 2D triangular lattice. By accessing the momentum dependence, our results shed light on a much richer physics of electromagnons that is beyond the reach of THz light experiments. The reported measurement also shows how inelastic X-ray scattering can be used to probe magnetic correlations with high energy and momentum resolution in certain systems with large enough magnon–phonon coupling. This study will open a route towards measuring magnetic correlations at extreme conditions using diamond anvil cells. Indeed, IXS can be performed with samples as thin as 10–20 μm, which allow extending such studies up to Mbar pressure[38]. It is furthermore possible to work with evanescent wave fields in grazing angle conditions, which allows surface sensitive studies, measurements on thin films and multilayer systems[39,40].

## Methods

**Crystal growth.** $LiCrO_2$ single crystals were grown by the $Li_2O$–$B_2O_3$ flux or $Li_2O$–$PbO$–$B_2O_3$ flux methods for IXS and INS measurements, respectively. A typical growth was done by a mixture of $Li_2O$, $Cr_2O_3$ and $B_2O_3$ or with additional PbO. The mixture was heated at 1,300 °C and then slowly cooled down to 800 or 900 °C, respectively.

**Inelastic X-ray scattering.** IXS was measured on the ID28 beamline at the European Synchrotron Radiation Facility along the reciprocal space direction ($h$, $h$, 0) at temperatures 295, 99, 69, 40 and 7 K using incident photon energy of 17.794 keV ($\lambda = 0.6968$ Å) produced by the (9, 9, 9) Si Bragg reflection and beam size of $50 \times 50$ μm$^2$. Since the sample was a thin plate perpendicular to (0, 0, 1), we choose the ($h$, $h$, $l$) scattering plane to minimize absorption. The ID28 instrumental energy resolution has a pseudo-Voigt profile with 2.71(2) and 3.3(1) meV full width at half maximum of the Gaussian and Lorentzian components and a mixing parameter of 0.63(2). The momentum resolution of the ID28 spectrometer is close to rectangular with 0.027 and 0.076 Å$^{-1}$ horizontal and vertical width perpendicular to the momentum transfer, while the longitudinal momentum resolution is at least two orders of magnitude better than the transverse.

**Inelastic neutron scattering.** INS was measured on the EIGER triple-axis spectrometer at SINQ at the Paul Scherrer Institut using fixed final neutron energy of 14.7 meV, double focusing graphite monochromator and horizontal focusing graphite analyser. To eliminate spurious scattering a pyrolytic graphite filter was applied after the sample. We have used a 50 mg single crystal of $LiCrO_2$ and performed measurements at 1.5 K. Owing to the small sample size, the spin wave signal was only collected close to the magnetic Bragg points along the ($h$, 1–2$h$, 0) direction and at (1/2, 1/2, 0) in reciprocal space. The spin wave peak as a function of neutron energy transfer was fitted with a Gaussian function.

**Curve fitting.** All constant momentum transfer scans were fitted with a line shape that is the instrumental energy resolution convoluted with a Lorentzian to model the finite lifetime of the excitations. In the main text all intrinsic line width are given by the full width at half maximum of the Lorentzian component. All given error are 1 s.d., originating from the statistical error of the detector counts.

**Phonon calculation.** Lattice dynamics calculations were performed using the finite displacement method within density functional theory (DFT)[41]. Distorted atomic configurations were generated and the induced forces of a $4 \times 4 \times 4$ supercell were computed by total energy calculations using Projector Augmented Waves method as implemented in VASP[42–44]. A shifted $4 \times 4 \times 4$ momentum grid is used for the ionic relaxations and the calculation of Born Effective charges by perturbation theory[45] in the primitive unit cell. While the internal ionic coordinates are relaxed, the lattice constant is kept fixed to the experimental value to reduce the error due to unit cell volume. The valence electrons are treated explicitly by the VASP PAW potentials are $1s^2 2s^2 sp^1$ for Li, $3p^6 3d^5 4s^1$ for Cr and $2s^2 2p^4$ for O. A plane wave cutoff of 500 eV, which is 25% larger than suggested, is used and tested to provide good convergence. PBEsol exchange correlation functional[46,47] is used for all calculations. To account for the underestimation of on-site correlations by generalized gradient approximation (GGA), the DFT + $U$ approximation[48] is used with a $U$ of 3 eV, which has previously been shown to faithfully reproduce the spin-phonon properties of Cr oxides in the same implementation[49]. Dynamical matrices throughout the Brillouin zone were computed using Fourier transformation as implemented in Phonopy[50] and non-analytical term corrections due to finite Born charges were applied. A shifted $4 \times 4 \times 4$ momentum grid has been used for sampling the electronic structure of the primitive unit cell.

**Model of the magnon–phonon coupling.** The spin wave model and the coupled magnon–phonon model was solved numerically using a modified version of SpinW (ref. 32).

**Data availability.** All relevant data that support the findings of this study are available from the corresponding author on request.

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

## Acknowledgements

We thank Andrea Scaramucci and Michel Kenzelmann for helpful discussions on electromagnons. The IXS measurements were performed on beamline ID28 at the European Synchrotron Radiation Facility, Grenoble, France. We are grateful to Thanh-Tra for providing assistance in using beamline ID28 and Christina Drathen for collecting data on beamline ID22. We also thank Céline Besnard for preliminary X-ray diffraction measurements performed at University of Geneva. Part of this work is based on experiments performed at the Swiss spallation neutron source SINQ, Paul Scherrer Institute, Villigen, Switzerland. The research leading to these results has received funding from the European Community's Seventh Framework Programme (FP7/2007–2013) under Grant Agreement No. 290605 (COFUND: PSI-FELLOW). K.K. and T.K. were supported by JSPS KAKENHI (grant no. 24244058).

## Author contributions

S.T. and B.W. carried out inelastic X-ray spectroscopy and analysed data; S.T., K.R. and U.S. carried out INS experiments and analysed data; K.R., H.T., K.K. and T.K. synthesized samples; B.W. and T.B. carried out *ab initio* calculations; the results were discussed and interpreted by S.T. and C.R.; the manuscript was written by S.T. with input from all the authors.

## Additional information

**Competing financial interests:** The authors declare no competing financial interests.

