## [Peer Review File · Nature Communications]

Reviewers' comments:

Reviewer #1 (Remarks to the Author):

Toth et al. reports an observation of electromagnon dispersion in LiCrO₂ using inelastic x-ray scattering (IXS), which is usually only sensitive to phonons. Through strong phonon-magnon mixing, it is claimed that electromagnons become visible in IXS spectra in a non-resonant condition. The authors also provide ab initio calculations of the phonon modes and a theoretical model accounting for their data based on a linear coupling of phonon and magnon due to exchange striction. The authors have made a rather extensive study on the material and the conclusions they draw are generally agreeable. However, I find that the paper targets too narrow audience and is only comprehensible to experts in the field. I did not find anything particular that would be of broad interest. Therefore, I recommend publication in a more specialized journal. Below I list some of the questions/comments which authors may find useful.

- (1) Electromagnons have been seen a number of times previously and it is not surprising that they would be visible to any probe sensitive to phonons.
- (2) That the 10 meV mode is visible to neutron does not necessarily imply a magnetic character of the mode. It would be helpful to show q-dependence and/or polarization analysis of the mode.
- (3) The spectra near the magnetic Bragg peak show broad continuum, which authors explain in terms of coupling of phonon to two-magnons. This statement is not substantiated. The authors use spin-only models to show that two-magnons have large spectral weights near the ordering wave vector. Even if the spin-only model was valid, one would still expect to see the dominant single magnon peak, which is not the case. Moreover, this model does not explain the abrupt change seen between spectra in panels (h) and (i) of Fig. 3 shown for q points far away from the ordering wave vector.
- (4) The authors provide an explanation for why only one spin-wave mode out of possible three is observed. I find this explanation a bit terse and probably this can be made more accessible especially for non-experts.
- (5) Probably the analysis of the temperature dependence (Fig. S5) belongs in the main text.
- (6) The summary paragraph can be made more concise. It tries to touch on too many things that are not of direct relevance to this paper and do not add any value to the paper. (e.g. high pressure, surface sensitive measurements, thin films..)

Reviewer #2 (Remarks to the Author):

The authors report inelastic x-ray and neutron scattering measurements on the triangular lattice antiferromagnet LiCrO₂, which show the appearance of a sharp excitation at low temperatures with characteristics of both a phonon and a spin wave. The authors map out at least some of the dispersion and spectral weight for both this "electromagnet mode" as well as more conventional phonons and two magnon contributions. This is an interesting paper that describes quite sophisticated x-ray and neutron experiments and the authors have done due diligence in explaining what this new excitation is likely to be, and how it arises.

I have several queries for the authors. However, provided that they respond reasonably to these queries, I think it is likely that I would recommend publication of this manuscript in Nature Communications.

1- The authors have some information regarding the crystallography of their single crystals in the supplemental material. However I found it surprising that there is no mention of the fact that Li^+ and Cr^{3+} can have very similar ionic radii (at least for the coordination of Li^+ that is relevant) and it may be quite easy to have Cr on the Li site and vice-versa. This is known to occur for Li^+ and Ni^{3+} in LiNiO_2 , for example, at the 1-5 % level and may be responsible for the suppression of T_n in that case (see, for example, Lewis et al, PRB Phys. Rev. B, 72, 014408 1-5)

If some Li - Cr mixing is present, could such a magnetic defect in the structure provide a possible coupling between phonons and magnons, with some of the properties described here?

2- On page 1 the authors write " The electric field component of the light at the resonant frequency can excite and measure the electromagnets". I understand what it means to excite an excitation - what does "measure" an excitation mean? This is somewhat vague, and it a better choice of wording may improve the clarity.

3- Also on page 1 the authors refer to the interplane interactions as being weak, but then say that they give rise to a relatively high $T_n \sim 61.2$ K. As it is these interplane interactions that drive the phase transition, they do not sound like they are all that weak.

4- On page 3 the authors refer to the vestige of the electromagnon above T_n as a PEM "which persist far above T_n due to the low dimensionality of the system". However the data at 69 K show the PEM to be quite broad in energy , even though the system is at the edge of its conventional 3D critical regime ($T-T_n/T_n \sim 0.1$). At 99 K, it is completely washed out. Consequently I see little evidence for the PEM surviving with a long lifetime (energy width) to temperatures well above T_n , as might be expected if the critical regime were anomalously large (as it is for low dimensional systems).

5- Also on page 3, the authors mention that "the sum of the spectral weight of the lowest phonon mode and the low temperature resonance is independent of temperature, .. " This appears to be roughly correct from inspection - however should it not be the imaginary part of the dynamic susceptibility, $\chi''(Q, \omega)$, that would be conserved with such an argument - not $S(Q, \omega)$? Fig. 2 shows data from $T=295$ K to 7 K, and the modes in question are at ~ 30 meV and 10 meV, so the Bose factor part of $S(Q, \omega)$ is changing a lot over that temperature range. Also, even for the case of a reasonably isolated optic phonon, such as the one near 60 meV, there does not appear to be a conservation law associated with the spectral weight of this mode.

Spectral weight does seem to be transferred from the LA phonon at high T to the EM at low T, but it may be better to simply say that and not claim that there is a hard conservation of the spectral weight between the two at this particular wave vector.

6- The authors draw attention to the "roton minimum in the spin wave dispersion at K downwards renormalizing the phason energy by 42% .. ". In believe the authors are referring here to the fact that the low energy inelastic feature at ~ 10 meV at (1.5, 1.5, 0) departs from the parabolic, calculated spin wave dispersion by $\sim 40\%$ of the calculated spin wave energy at (1.5, 1.5, 0).

I think this is an interesting observation, but the measured dispersion of this mode is actually very flat around (1.5 1.5 0) as shown in Fig. 3. It is not clear to me that it softens at all at (1.5 1.5 0) and I don't think an analogy to the roton in liquid 4He is helpful (in the real roton case the excitation really

does soften in energy around the roton wavevector.) I believe the readership will find this confusing, as they may be looking for an excitation whose energy really softens - not simply that its dispersion is not as predicted by a calculation.

7- While polarization analysis of the IXS experiment was not performed, the polarized nature of synchrotron radiation may be such that certain sensitivities to polarization may still be present in a nominally unpolarized IXS experiment. Is this the case, and is the electromagnon expected to have specific characteristics which could be elucidated with "full polarization analysis" of the IXS experiment?

8- The rationale for the choice of the 17.794 keV incident x-ray beam for the IXS experiment is not given. What was this rationale? The authors explain that for their analysis it is important that the IXS experiment be off-resonance, but (generally) could there be an advantage to tuning to an appropriate absorption edge?

Reviewer #3 (Remarks to the Author):

This is a primarily experimental paper, devoted to excitation modes in the antiferromagnet LiCrO₂. It reports results from non-resonant inelastic x-ray scattering (IXS), a technique sensitive to phonon excitations. The authors argue that, due to strong magneto-elastic coupling, LiCrO₂ displays coupled magnon-phonon modes which are visible to IXS -- this is signified by a shift of spectral weight upon cooling from a bare phonon mode at elevated energy to a new mode at much lower energy which becomes sharp below the magnetic ordering temperature. The paper also contains a comprehensive theoretical modelling of the zero-temperature excitations, which supports the notion of strongly coupled electromagnon modes.

After having studied the paper, I am impressed by both the data and the agreement between experiment and theory. The authors' modelling appears to capture all salient features of the data, including the proper momentum-dependent hybridization of magnons and phonons and the resulting roton-like minimum of the lowest-energy mode. In the following, I list a few questions and remarks which the authors should take into account in a revised version.

(i) The authors state that the intensity of the LA phonon mode at higher temperatures is strongly underestimated by the ab-initio calculation, but do not discuss possible reasons. Does this point to a weakness in the modelling, to unknown matrix-element effects, or to something else?

(ii) The authors neglect interplane coupling entirely. Is there an estimate for its value to justify this neglect?

(iii) Below Eq (5) the text states "The linear coupling vanishes for non-collinear magnetic order, because ...". To be consistent with the argumentation, I believe this should read "collinear magnetic order".

(iv) The middle part of the first text paragraph reads a bit incoherent. I presume the authors want to motivate the use of IXS, but the logical flow of thoughts is hard to follow. This should be reformulated. Also, the use of the terms "majority component" and "minority component" in the abstract and the first paragraph is difficult to comprehend for non-specialists.

(v) The model calculation presented in the supplement works with rotated spin operators in Eq (S10) and (S11), however the same symbol, $\{\mathbf{S}\}_n$, as in Eq (1) is used (where it refers to spin

operators in the laboratory frame). To avoid confusion, I suggest to modify the notation (i.e. use a different symbol for rotated spins).

(vi) The "two-magnon correlation function" is shown in Figs 3(j) and S6, but not properly defined in the paper. I guess it would make sense to quote Eq (A6) from Coldea et al (which I guess is what the calculation is based upon), together with a few additional explanatory sentences.

In summary, this is a high-level paper with results relevant for the community working on complex magnets and multiferroics. Its combined experimental and theoretical results form the basis for further investigations, such as the physics in an external magnetic field. It also paves the way for using IXS as a general tool to measure magnetic excitations. Taken together, I believe that the paper reaches the level of importance required for a publication in Nature Communications.

Reviewers' comments:

Reviewer #1 (Remarks to the Author):

Toth et al. reports an observation of electromagnon dispersion in LiCrO₂ using inelastic x-ray scattering (IXS), which is usually only sensitive to phonons. Through strong phonon-magnon mixing, it is claimed that electromagnons become visible in IXS spectra in a non-resonant condition. The authors also provide ab initio calculations of the phonon modes and a theoretical model accounting for their data based on a linear coupling of phonon and magnon due to exchange striction. The authors have made a rather extensive study on the material and the conclusions they draw are generally agreeable. However, I find that the paper targets too narrow audience and is only comprehensible to experts in the field. I did not find anything particular that would be of broad interest. Therefore, I recommend publication in a more specialized journal. Below I list some of the questions/comments which authors may find useful.

We are glad that the referee finds the scientific content generally agreeable and we hope that our answers below would convince him/her about the broad scientific impact of our paper.

(1) Electromagnons have been seen a number of times previously and it is not surprising that they would be visible to any probe sensitive to phonons.

We agree with the referee. The fact that electromagnons can be observed through phonon sensitive probes is not new. However, detailed spectroscopic studies to date have been restricted to THz light absorption, which probes only the center of the Brillouin zone. As our results show, in case of sufficient magnon-phonon mixing non-resonant inelastic x-ray scattering can be used to probe electromagnon physics throughout the Brillouin zone that provides complete spectroscopic information for dispersive excitations. Moreover, using non-resonant x-ray scattering to study strongly coupled materials enables not only to probe the electromagnon dispersion, but to study correlated magnetism on systems where inelastic neutron scattering is impractical due to small sample sizes.

(2) That the 10 meV mode is visible to neutron does not necessarily imply a magnetic character of the mode. It would be helpful to show q-dependence and/or polarization analysis of the mode.

The magnetic character of our electromagnon mode can be established based on our inelastic neutron scattering (INS) data. The INS measurements were done at low Q (the scan at (1/2,0,0) is at $|Q| = 1.25 \text{ \AA}^{-1}$) where scattering from phonons is negligible (the intensity being $|Q|^2$ dependent), while the magnetic scattering is strong since the Q is well below the magnetic form factor cutoff of the Cr³⁺ ion (see figure below).

We measured the q-dependence of the dispersion, which is denoted in Fig. 1(c) by blue squares, moreover individual INS scans are shown in Fig. S2. Unfortunately, due to the small crystal size (50 mg) the study of the spin wave dispersion is restricted to regions that are close to the magnetic Bragg peaks and to low energy transfer where the scattering intensity is large. Due to the same reason a polarized neutron study as proposed by the referee is not possible a sample of this size. We note that we present theoretical modeling describing mode assignment and relative intensities very accurately for both X-ray and neutron data and for all phonon and spin wave modes.

(3) The spectra near the magnetic Bragg peak show broad continuum, which authors explain in terms of coupling of phonon to two-magnons. This statement is not substantiated. The authors use spin-only models to show that two-magnons have large spectral weights near the ordering wave vector. Even if the spin-only model was valid, one would still expect to see the dominant single magnon peak, which is not the case. Moreover, this model does not explain the abrupt change seen between spectra in panels (h) and (i) of Fig. 3 shown for q points far away from the ordering wave vector.

Based solely on the spin dynamic structure factor, one expects strong one magnon scattering close to the magnetic Bragg peaks. However due to the q-dependent coupling between the magnons and phonons the 1-magnon cross section for inelastic x-ray scattering (IXS) at the magnetic Bragg peaks is exactly zero. This enables the observation of the 2-magnon continuum that is generally much weaker than the 1-magnon cross section. The abrupt change visible between Fig. 3 (h) and (i) is due to the IXS intensity loss because of the aforementioned q-dependent magnon-phonon coupling.

(4) The authors provide an explanation for why only one spin-wave mode out of possible three is observed. I find this explanation a bit terse and probably this can be made more accessible especially for non-experts.

We are very thankful to the referee for this suggestion. We have included a much clearer explanation in the revised manuscript:

“The exchange interactions in the system can be thought of as effective magnetic fields acting on each magnetic site and being equal to the sum of the neighbouring moments times the exchange constant J_1 . In the absence of phonons, the field is parallel to the moment direction on every site as illustrated by purple arrows in Fig. 1(a). However, when a phonon perturbs the system and modulates the uniform J_1 via exchange striction, the effective magnetic field will not be parallel to the moment direction any more but points somewhere within the plane of the spin spiral. This will induce a modulation of the phase of the spins within the spiral as spins reorient themselves to minimise the total energy. This phase modulation is exactly the phason spin wave mode that we see in our data.”

(5) Probably the analysis of the temperature dependence (Fig. S5) belongs in the main text.

We thank the referee for this suggestion. We have included the figure in the main manuscript and adapted the wording in the text, emphasizing that it shows the imaginary part of the phonon susceptibility.

(6) The summary paragraph can be made more concise. It tries to touch on too many things that are not of direct relevance to this paper and do not add any value to the paper. (e.g. high pressure, surface sensitive measurements, thin films..)

Beyond the specific achievement of our study, these are the possibilities which make our current study of such broad interest: the ability of non-resonant IXS to address

electromagnon physics under conditions hitherto hardly accessible opens the door to many exciting studies. We consider it important to emphasize these future avenues in the summary. We note that other referees pointed out that our study paves the way for using IXS as a general tool to measure magnetic excitations in a wide range of systems.

Reviewer #2 (Remarks to the Author):

The authors report inelastic x-ray and neutron scattering measurements on the triangular lattice antiferromagnet LiCrO₂, which show the appearance of a sharp excitation at low temperatures with characteristics of both a phonon and a spin wave. The authors map out at least some of the dispersion and spectral weight for both this "electromagnet mode" as well as more conventional phonons and two magnon contributions. This is an interesting paper that describes quite sophisticated x-ray and neutron experiments and the authors have done due diligence in explaining what this new excitation is likely to be, and how it arises.

I have several queries for the authors. However, provided that they respond reasonably to these queries, I think it is likely that I would recommend publication of this manuscript in Nature Communications.

We thank the referee for finding our paper interesting and being positive about publication in Nature Communications. We answer all remaining questions below.

1- The authors have some information regarding the crystallography of their single crystals in the supplemental material. However I found it surprising that there is no mention of the fact that Li⁺ and Cr³⁺ can have very similar ionic radii (at least for the coordination of Li⁺ that is relevant) and it may be quite easy to have Cr on the Li site and vice-versa. This is known to occur for Li⁺ and Ni³⁺ in LiNiO₂, for example, at the 1-5 % level and may be responsible for the suppression of T_n in that case (see, for example, Lewis et al, PRB Phys. Rev. B, 72, 014408 1-5)

If some Li - Cr mixing is present, could such a magnetic defect in the structure provide a possible coupling between phonons and magnons, with some of the properties described here?

We thank the referee for this question. We measured x-ray powder diffraction on a LiCrO₂ powder sample to determine the precise low temperature crystal structure at 10 K that is shown in Supplementary Figure S1. Our refinement shows some oxygen deficiency (<5%) and occupation of more than 98% of both Li and Cr sites (relative precision of about 1%). Due to the strong x-ray absorption of the sample, site occupancy cannot be determined more precisely, as it strongly depends on the applied absorption correction. A powder neutron diffraction study might be able to quantify site occupancies more precisely in the future. But such an experiment has to be done on a large powder sample from several batches.

On the other hand, even if small site disorder is present, we don't see how it could introduce the observed coupling. Besides there is no need for such a defect-mediated coupling. Our magnon-phonon coupling model is based on the exchange striction mechanism that was found before in compounds with similar Cr-Cr distance and other systems having similar values to our J_{MP} (such as ZnCr₂O₄, see references in our manuscript).

To conclude, there is no evidence for significant site disorder, and even if there was, it is unlikely to influence the discussed magnon-phonon coupling. Therefore we do not think it is of value to a general audience to discuss this possibility.

2- On page 1 the authors write " The electric field component of the light at the resonant frequency can excite and measure the electromagnets". I understand what it means to excite an excitation - what does "measure" an excitation mean? This is somewhat vague, and it a better choice of wording may improve the clarity.

We thank the referee for pointing out this formulation. We improved the readability by removing the words "and measure".

3- Also on page 1 the authors refer to the interplane interactions as being weak, but then say that they give rise to a relatively high $T_N \sim 61.2$ K. As it is these interplane interactions that drive the phase transition, they do not sound like they are all that weak.

We do not have direct evidence on the size of the interplane interactions. However LiCrO_2 can be compared to $\alpha\text{-CaCr}_2\text{O}_4$ (which has similar crystal structure plus a small orthorhombic distortion), see the paper S. Toth et al., PRL 109, 127203 (2012). In $\alpha\text{-CaCr}_2\text{O}_4$ the interplane coupling of 0.027 meV drives T_N to 42.6 K. The reason for the high T_N is that it depends not only on the weak interplane coupling but also the strong in-plane couplings. In low dimensional magnets ordering temperatures are generally higher than the energy scale of the weak couplings between strongly correlated units.

4- On page 3 the authors refer to the vestige of the electromagnon above T_N as a PEM "which persist far above T_N due to the low dimensionality of the system". However the data at 69 K show the PEM to be quite broad in energy, even though the system is at the edge of its conventional 3D critical regime ($T-T_N/T_N \sim 0.1$). At 99 K, it is completely washed out. Consequently I see little evidence for the PEM surviving with a long lifetime (energy width) to temperatures well above T_N , as might be expected if the critical regime were anomalously large (as it is for low dimensional systems).

We agree with the referee that the supposed PEM excitation does not survive as a well-defined mode at temperatures well above T_N and more precise temperature and momentum dependent study is necessary to unambiguously identify it. Thus we changed the corresponding text and removed the word paraelectromagnon from the introduction and the conclusion:

"Moreover, at intermediate temperatures above T_N the strongly damped low energy excitation *might be* a paraelectromagnon (PEM) due to the lack of both magnetic and electric dipole order. *The origin of this excitation can be* a phonon coupled to the excitation of the 2D correlated magnetic state, which persists ~~far~~ above T_N due to the low dimensionality of the system."

5- Also on page 3, the authors mention that "the sum of the spectral weight of the lowest phonon mode and the low temperature resonance is independent of temperature, .." This appears to be roughly correct from inspection - however should it not be the imaginary part of the dynamic susceptibility, $X''(Q, \omega)$, that would be conserved with such an argument - not $S(Q, \omega)$? Fig. 2 shows data from $T=295$ K to 7 K, and the modes in question are at ~ 30 meV and 10 meV, so the Bose factor part of $S(Q, \omega)$ is changing a lot over that temperature range. Also, even for the case of a reasonably isolated optic phonon, such as the one near 60 meV, there does not appear to be a conservation law associated with the spectral weight of this mode.

Spectral weight does seem to be transferred from the LA phonon at high T to the EM at low T , but it may be better to simply say that and not claim that there is a hard conservation of the spectral weight between the two at this particular wave vector.

We thank the referee for this question. We made our statement more precise in the text. We indeed discuss the temperature dependence of the imaginary part of the dynamic susceptibility $X''(Q, \omega)$. We moved Fig. S5 to the main text and improved the labels. What we previously called the “intensity corrected for the Bose-factor” meant the imaginary part of the dynamic susceptibility. The change of the intensity of the optical phonon is quite well described by the change of the Debye-Waller factor. This can be seen in Fig. 2, where our model predicts the optical phonon intensity well both at 7 K and at 295 K simply by changing the Debye-Waller factor (there is no arbitrary scaling between the scans). Also our magnon-phonon coupled model gives a conservation of the sum of X'' of the coupled phonon and magnon as we vary J_{MP} . Thus X''_1 of the decoupled phonon when $J_{MP} = 0$ (which is equivalent to the $T = 295$ K measurement) is equal to the measured $X''_{EM} + X''_{LA}$ at low temperature (finite J_{MP}).

6- The authors draw attention to the "roton minimum in the spin wave dispersion at K downwards renormalizing the phason energy by 42% .. ". In believe the authors are referring here to the fact that the low energy inelastic feature at ~ 10 meV at $(1.5, 1.5, 0)$ departs from the parabolic, calculated spin wave dispersion by $\sim 40\%$ of the calculated spin wave energy at $(1.5, 1.5, 0)$.

I think this is an interesting observation, but the measured dispersion of this mode is actually very flat around $(1.5, 1.5, 0)$ as shown in Fig. 3. It is not clear to me that it softens at all at $(1.5, 1.5, 0)$ and I don't think an analogy to the roton in liquid ^4He is helpful (in the real roton case the excitation really does soften in energy around the roton wavevector.) I believe the readership will find this confusing, as they may be looking for an excitation whose energy really softens - not simply that its dispersion is not as predicted by a calculation.

The expression “roton minimum” or “roton like minimum” is widely used in the literature of the triangular lattice antiferromagnet (TLA) denoting the weakly dispersive M' point (we exchanged the labels M and K in the corrected manuscript to follow the common practice of identifying BZ corner (K) and edge middle (M) points) in reciprocal space due to quantum fluctuations. One of the first mention of it was in the paper W. Zheng, et al., PRL 96, 057201 (2006) and it can be found in several subsequent papers. Although we are aware of the missing connection with the roton mode in superfluid helium, we think the “roton minimum” expression identifies an important point (edge center) of the Brillouin zone of the triangular lattice antiferromagnet.

Since there is no relation to ^4He physics, the softening of the magnon energy at the M' should not be compared to the softening of the roton in ^4He as a function of temperature. In our case the softening (or one may call it ‘renormalization’) of the magnon energy is due to the magnon-phonon interaction (thus present at all temperatures where magnons exist). Although the local minima of the magnon dispersion at the roton point is not extremely expressed, it is a large downwards deviation from the pure magnon energy.

In conclusion, while keeping in mind that there is no connection to ^4He physics, we would use the expression “roton-like minimum” in the manuscript to relate to previous studies of this high symmetry point of the Brillouin zone.

7- While polarization analysis of the IXS experiment was not performed, the polarized nature of synchrotron radiation may be such that certain sensitivities to polarization may still be present in a nominally unpolarized IXS experiment. Is this the case, and is the electromagnon expected to have specific characteristics which could be elucidated with "full polarization analysis" of the IXS experiment?

The incident x-ray beam is polarized due to the nature of the source, however the polarization is not analyzed in the detector. Since there is no additional selection rule for phonons if the x-ray beam is polarized, polarization analysis would not provide additional information on the phonon spectrum. On the other hand, polarization analysis can differentiate between charge scattering (phonons) and magnetic scattering. But due to the small cross section of magnetic scattering (4×10^{-6} weaker in the keV range than charge scattering, for details see E. Burkel, Rep. Prog. Phys. 63, 171 (2000)), it is not detectable in our experimental setup.

8- The rationale for the choice of the 17.794 keV incident x-ray beam for the IXS experiment is not given. What was this rationale? The authors explain that for their analysis it is important that the IXS experiment be off-resonance, but (generally) could there be an advantage to tuning to an appropriate absorption edge?

The x-ray energy of our IXS measurement is chosen by the specific Bragg reflection of the Si-crystal analyzer, the (9,9,9) reflection in our case. The crystal analyzer and backscattering geometry is necessary to reach meV resolution which works only for hard x-rays. Unfortunately, this energy cannot be tuned to measure an arbitrary absorption edge where the signal of a specific type of atom could be amplified. On the other hand, tunable RIXS spectrometers use gratings that do not have the necessary energy resolution for our type of study (compare the 30-100 meV RIXS energy resolution on the Cu L-edge to the 1-3 meV energy resolution of IXS). This makes our study interesting, since up to now studies of magnons by x-rays were only possible on RIXS instruments with poor energy resolution. We show for the first time that magnetic excitations can be measured on high energy resolution IXS spectrometers if the system has a sufficiently strong spin-phonon coupling.

A further challenge for RIXS could be that typical resonance energies for 3d transition metal ions and oxygen are in the soft x-ray region where the x-ray photons do not have the necessary momentum to study the complete Brillouin zone.

Reviewer #3 (Remarks to the Author):

This is a primarily experimental paper, devoted to excitation modes in the antiferromagnet LiCrO₂. It reports results from non-resonant inelastic x-ray scattering (IXS), a technique sensitive to phonon excitations. The authors argue that, due to strong magneto-elastic coupling, LiCrO₂ displays coupled magnon-phonon modes which are visible to IXS -- this is signified by a shift of spectral weight upon cooling from a bare phonon mode at elevated energy to a new mode at much lower energy which becomes sharp below the magnetic ordering temperature. The paper also contains a comprehensive theoretical modelling of the zero-temperature excitations, which supports the notion of strongly coupled electromagnon modes.

After having studied the paper, I am impressed by both the data and the agreement between experiment and theory. The authors' modelling appears to capture all salient features of the data, including the proper momentum-dependent hybridization of magnons and phonons and the resulting roton-like minimum of the lowest-energy mode. In the following, I list a few questions and remarks which the authors should take into account in a revised version.

We thank the referee for his/her kind opinion on our manuscript and for emphasizing its high level of importance and impact necessary for a publication in Nature Communications.

(i) The authors state that the intensity of the LA phonon mode at higher temperatures is

strongly underestimated by the ab-initio calculation, but do not discuss possible reasons. Does this point to a weakness in the modelling, to unknown matrix-element effects, or to something else?

This is an interesting point. To our best knowledge the discrepancy between measured phonon energies and the *ab-initio* calculation is due to one of the systematic errors in DFT. It could be:

- 1) An error due to the exchange correlation functional or the DFT+U approximation. We disregard both nonlocal exchange and correlations.
- 2) An error because of the way we treat magnetism. In the phonon calculations, we disregard not only the spin-orbit coupling (which is likely to be small) but also magnetic correlations that may persist at temperatures above T_N .
- 3) A problem due to the quantum fluctuations of the positions of the nuclei. Li is quite light and might require special treatment for the quantum effects in its oscillations. This is a point that is particularly beyond DFT.

We hope this question will be elucidated in the future.

Since this is mainly an experimental paper and our ab-initio results reproduce the experimental phonon dispersions relatively well apart from the speed of sound, we do not think it is of value for the general audience to discuss the source of systematic errors in the calculation.

(ii) The authors neglect interplane coupling entirely. Is there an estimate for its value to justify this neglect?

The value of the interplane coupling in LiCrO_2 is not known. However the electromagnon dispersion along the measured $(h,h,0)$ direction is insensitive to the interplane coupling at low temperature, thus we do not need to make any assumption on its size to model our data. It would most probably introduce a small dispersion along the $(1/3,1/3,l)$ -direction where the in-plane magnon energy is small.

Also it is important to note that the interplane couplings are also frustrated due to the rhombohedral stacking of the triangular layers. Thus their value cannot be estimated simply from the ordering temperature and J_1 , see our similar answer above. We can point to a similar compound $\alpha\text{-CaCr}_2\text{O}_4$ (which has similar crystal structure as LiCrO_2 plus a small orthorhombic distortion, see the paper S. Toth et al., PRL 109, 127203 (2012)) where the interplane coupling is 0.027 meV and less than 1% of J_1 .

(iii) Below Eq (5) the text states "The linear coupling vanishes for non-collinear magnetic order, because ...". To be consistent with the argumentation, I believe this should read "collinear magnetic order".

We thank the referee for pointing this out. We corrected the mistake.

(iv) The middle part of the first text paragraph reads a bit incoherent. I presume the authors want to motivate the use of IXS, but the logical flow of thoughts is hard to follow. This should be reformulated. Also, the use of the terms "majority component" and "minority component" in the abstract and the first paragraph is difficult to comprehend for non-specialists.

We removed the expressions "majority component" and "minority component" from the text and rewrote the first paragraph to make it more clear and easier to read:

"The coupling between magnetic and lattice degrees of freedom gives rise to many interesting effects. It can induce multiferroic order with ferroelectric polarisation coupled to the magnetic structure [1–3] or it can generate dynamic mixed magnon–

phonon excitations. If the magnon is coupled to a polar phonon, the mixed mode, termed electromagnon, can be excited by the electric field of light at the resonant frequency [4-6]. Previous experiments showed that magnetization dynamics of materials can be studied at ultrafast time scales by exciting electromagnons via femtosecond light pulses [7]. Moreover, optical properties of solid state materials at the electromagnon resonance can be controlled via external magnetic field [8,9]. Measurement of electromagnons is possible via THz spectroscopy. However, this technique is able to probe only the center of the Brillouin zone. As we shall show, electromagnons can appear at finite momentum, inaccessible to THz spectroscopy. Inelastic neutron scattering can also identify the magnetic and phononic component of an electromagnon excitation, however previous studies found only small energy shifts of the magnons due to magnon–phonon coupling [10,11] while **the transfer of spectral weight between magnons and phonons could not be resolved so far**. Here we show that LiCrO₂ is an exceptional material where the magnon–phonon coupling is strong enough to make **the transferred spectral weight from phonons to magnons visible** for inelastic x-ray scattering and thus enables the direct measurement of the electromagnon dispersion.”

(v) The model calculation presented in the supplement works with rotated spin operators in Eq (S10) and (S11), however the same symbol, $\{\mathbf{S}\}_n$, as in Eq (1) is used (where it refers to spin operators in the laboratory frame). To avoid confusion, I suggest to modify the notation (i.e. use a different symbol for rotated spins).

We thank the referee for pointing this out. We changed the labelling of the spin operators in the rotating frame by S' in the Supplementary Materials to avoid confusion.

(vi) The "two-magnon correlation function" is shown in Figs 3(j) and S6, but not properly defined in the paper. I guess it would make sense to quote Eq (A6) from Coldea et al (which I guess is what the calculation is based upon), together with a few additional explanatory sentences.

We included a short explanation to the two magnon continuum and slightly changed our wording, see the changed text in red italic below.

“Since the one magnon excitations are sharp at low temperature, the broad IXS peaks can be due to phonons coupled to the two-magnon (2M) continuum that is intrinsically broad for dispersive magnons. ***The two-magnon continuum is purely a quantum effect and the corresponding dynamical structure factor is typically much smaller than that of the single magnon. It is related to the longitudinal fluctuations of the ordered spins. \cite{Coldea2003} To corroborate our argument,*** we calculated the non-interacting two-magnon dynamical structure factor for the TLA ***\cite{Coldea2003}*** with first and second neighbour antiferromagnetic interactions $J_1 = 8.17$ meV and $J_2 = 0.556$ meV, shown in Fig. 4(j), and a cut at $(1.292, 1.292, 0)$ in reciprocal space is shown in Fig. 4(e). ***The two-magnon dynamical structure factor*** is strongest close to the M point and the centre of the two-magnon spectral weight is expected to be close to the one magnon energy.”

In summary, this is a high-level paper with results relevant for the community working on complex magnets and multiferroics. Its combined experimental and theoretical results form the basis for further investigations, such as the physics in an external magnetic field. It also paves the way for using IXS as a general tool to measure magnetic excitations. Taken together, I believe that the paper reaches the level of importance required for a publication in Nature Communications.

REVIEWERS' COMMENTS:

Reviewer #1 (Remarks to the Author):

I am satisfied with the answers from the authors and revisions made. This is a decent piece of work with high quality data and careful analysis. However, I still cannot agree that the paper has a broad enough impact to be published in Nature Communications. In particular, I strongly disagree with the Reviewer #3's comment that this work paves way for using IXS as a general tool to magnetic excitations. It only works for a handful of materials for almost a trivial reason--strong magnon-phonon coupling--which is an exceptional property of LiCrO₂ and other few related materials. Is then LiCrO₂ by itself of a broad interest to the community? Again, I agree that this work reveals quite a lot on the physical properties of LiCrO₂ by applying IXS for the first time on this material to study magnetic excitations, but I do not see innovative aspects at the level of Nature Communications.

Reviewer #2 (Remarks to the Author):

I have reviewed the new manuscript as well as the reports of all three referees and the responses of the co-authors. Overall, I feel the authors have done a reasonable job of responding to my comments, and to those of the other referees. As my original recommendation was generally positive towards publication in Nature Communications, once my comments were addressed, I feel that this manuscript is appropriate for publication in Nature Communications.

Reviewer #3 (Remarks to the Author):

The reply of the authors clarifies most of the issues raised in the first round of refereeing. The changes made to the manuscript are reasonable. As stated before, I consider this a paper which is worth being published in Nature Communications, and I support publication in its present form.

Reviewer #1 (Remarks to the Author):

I am satisfied with the answers from the authors and revisions made. This is a decent piece of work with high quality data and careful analysis. However, I still cannot agree that the paper has a broad enough impact to be published in Nature Communications. In particular, I strongly disagree with the Reviewer #3's comment that this work paves way for using IXS as a general tool to magnetic excitations. It only works for a handful of materials for almost a trivial reason--strong magnon-phonon coupling--which is an exceptional property of LiCrO₂ and other few related materials. Is then LiCrO₂ by itself of a broad interest to the community? Again, I agree that this work reveals quite a lot on the physical properties of LiCrO₂ by applying IXS for the first time on this material to study magnetic excitations, but I do not see innovative aspects at the level of Nature Communications.

We are glad that the referee is satisfied with all our answers. Regarding his comments on the lack of general interest of our results, we added an extra section to the discussion in our manuscript. Here we point out the limitations of the applicability of our technique to other compounds and also included a few examples where interesting physics can be revealed by IXS. We hope the referee is satisfied with these final comments.

The added section:

“There are potentially many other magnetic correlated systems where the magnon-phonon coupling is present and matrix elements are allowed by symmetry. However, for a measurable hybridization between magnetic and lattice fluctuations a large coupling is necessary that makes only a few of them suitable for studying magnetism via IXS. For example, ZnCr₂O₄ and MgCr₂O₄ with pyrochlore structures have exchange striction values comparable to LiCrO₂. It is also possible that the observed molecular resonance like magnetic signal in these systems is related to hybridised magnon phonon modes. Besides, large magnon phonon coupling is expected for magnetic 5*d* systems, where the extended *d* orbitals can support large modulation of the superexchange interaction due to ligand vibrations. For example, in NaOsO₃ an upward shift of the optical phonon energy by 5 meV was attributed to the onset of magnetic correlations. 5*d* systems are especially promising for IXS studies as the general lack of large single crystals prohibits detailed INS experiments.”

Reviewer #2 (Remarks to the Author):

I have reviewed the new manuscript as well as the reports of all three referees and the responses of the co-authors. Overall, I feel the authors have done a reasonable job of responding to my comments, and to those of the other referees. As my original recommendation was generally positive towards publication in Nature Communications, once my comments were addressed, I feel that this manuscript is appropriate for publication in Nature Communications.

Reviewer #3 (Remarks to the Author):

The reply of the authors clarifies most of the issues raised in the first round of refereeing. The changes made to the manuscript are reasonable. As stated before, I consider this a paper which is worth being published in Nature Communications, and I support publication in its present form.